# A unified strategy toward total syntheses of lindenane sesquiterpenoid [4 + 2] dimers

Biao Du[1], Zhengsong Huang[1], Xiao Wang[1], Ting Chen[1], Guo Shen[1], Shaomin Fu[1] & Bo Liu [1,2]

The dimeric lindenane sesquiterpenoids are mainly isolated from the plants of Chloranthaceae family. Structurally, they have a crowded molecular scaffold decorated with more than 11 stereogenic centers. Here we report divergent syntheses of eight dimeric lindenane sesquiterpenoids, shizukaols A, C, D, I, chlorajaponilide C, multistalide B, sarcandrolide J and sarglabolide I. In particular, we present a unified dimerization strategy utilizing a base-mediated thermal [4 + 2] cycloaddition between a common furyl diene, generated in situ, and various types of dienophiles. Accordingly, all the three types of lindenane [4 + 2] dimers with versatile biological activities are accessible, which would stimulate future probing of their pharmaceutical potential.

---

[1] Key Laboratory of Green Chemistry & Technology of Ministry of Education, College of Chemistry, Sichuan University, 610064 Chengdu, China. [2] State Key Laboratory of Natural Medicines, China Pharmaceutical University, 210009 Nanjing, China. Correspondence and requests for materials should be addressed to B.L. (email: chembliu@scu.edu.cn)

The dimeric lindenane sesquiterpenoids are a group of biologically active complex natural products mainly isolated from the plants of Chloranthaceae family. Structurally, these [4 + 2] dimers possess congested frameworks of at least eight rings decorated by more than 11 stereogenic centers[1–3]. Three types (types 1–3, Fig. 1a) can be categorized according to various substitution pattern on the ring B. Owing to their structural diversity and biological activities, numerous efforts have been made toward syntheses of the corresponding dimers or monomers[4–21]. Following our long-standing interest in the syntheses of terpenoids[22], we reported the first syntheses of lindenane sesquiterpenoid [4 + 2] dimers, sarcandrolide J and shizukaol D (type 2) in the guidance of our modified biosynthetic hypothesis (Fig. 1b)[6]. Recently, the Peng group reported their

**Fig. 1** Three types of natural lindenane [4 + 2] dimers and related biosynthetic hypothesis. **a** The dimeric lindenane sesquiterpenoids can be categorized into three types, among which type 3 dimers are superior in numbers. **b** The skeleton of the dimeric lindenane sesquiterpenoids can be constructed through a [4 + 2] cycloaddition between the common diene and different types of dienophiles

achievement of total syntheses of shizukaols A and E, types 1 and 2 [4 + 2] dimers, respectively, through the other modified biomimetic Diels–Alder reaction[7].

However, no synthesis of type 3 [4 + 2] dimers has yet been reported, despite their predominant numbers accounting for more than four-fifths across the whole lindenane family. Notably, direct functionalization of the *gem*-disubstituted alkene on ring B failed to convert type 1 dimers into types 2 or type 3, because the inherent stereochemical control from the robust 3/5/6/5 backbone favors the undesired diastereomer[6,16]. Thus, a unified synthetic strategy is in demand to provide all the three types of lindenane [4 + 2] dimers and their synthetic analogs, by following the plausible [4 + 2] biosynthetic pathways with a common furyl diene and various dienophiles.

Since the corresponding furyl diene is instable and not isolable, an acid-mediated diene-formation is pivotal to trigger the subsequent cycloaddition in our previous synthesis of type 2 [4 + 2] dimers (Fig. 2a)[6]. However, the *gem*-disubstituted alkene substructure on ring B of type 1 dimers may be vulnerable under acidic conditions, as the acid-promoted *exo/endo* cyclic alkene isomerization was observed, transforming bolivianine to isobolivianine (Fig. 2b)[23–25].

We conceive and realize a unified base-promoted strategy enabling divergent and biomimetic syntheses of all types of lindenane [4 + 2] dimers (Fig. 2c). Among this strategy, a type 3 lindenane dimer, sarglabolide I, could serve as a common synthetic precursor toward other type 3 dimers through versatile acetylations. Of note, the common diene precursor used in this strategy is more synthetically viable than that previously used in acid-promoted cycloaddition (12 steps vs. 17 steps).

## Results

**Synthesis of the diene precursor.** We initiated synthesis of the common diene precursor from a known Michael adduct **19** from verbenone (Fig. 3a)[23–25]. Oxidation to a 1,2-diketone and subsequent ring-opening of the cyclobutane in the presence of boron trifluoride afforded enone **20**. Then, a formal oxidative [3 + 2] cycloaddition in the presence of cerium ammonium nitrate (CAN), followed by treatment with Amberlsyt-15, was employed between compound **20** and silyl enol ether **21a** to afford adduct **22**[26,27]. Other single oxidants such as 2,3-Dichloro-5,6-dicyano-*p*-benzoquinone (DDQ) and Cerium(IV) sulfate only got the complex mixture. Intriguingly, acetonitrile/benzene (4:1) cosolvents serve as the optimal solvent system for the oxidative coupling step, whereas the individual solvent only delivered inferior results, probably due to compromised equilibrium between solubility of CAN and the solvent polarity. Moreover, application of **21b** and **21c** instead of **21a** in this reaction resulted in lower yields. Mechanistically, CAN may serve as the single electron oxidant to convert **21a** into the radical cation **I**, which could undergo a radical Michael addition to compound **20**[26]. The forming intermediate **II** would be further oxidized to the cationic intermediate **III**, which would be quenched by water to produce a mixture of aldehyde **IVa** and semi-acetal **IVb**. Subsequently, Amberlyst-15 promoted dehydration and facilitated aromatization to give furan **22** (Fig. 3b).

A chemo-selective allylic oxidation of **22** yielded an enal smoothly using selenium dioxide, while the furyl methylene and methyl remained untouched. The hydrazone formation followed by the Rh-catalyzed intramolecular cyclopropanation generated compound **23** with proper stereochemistry. In contrast, our original Pd-catalyzed cyclopropanation was proved unsuccessful[28]. Treating **23** with SeO$_2$ yielded **24** and

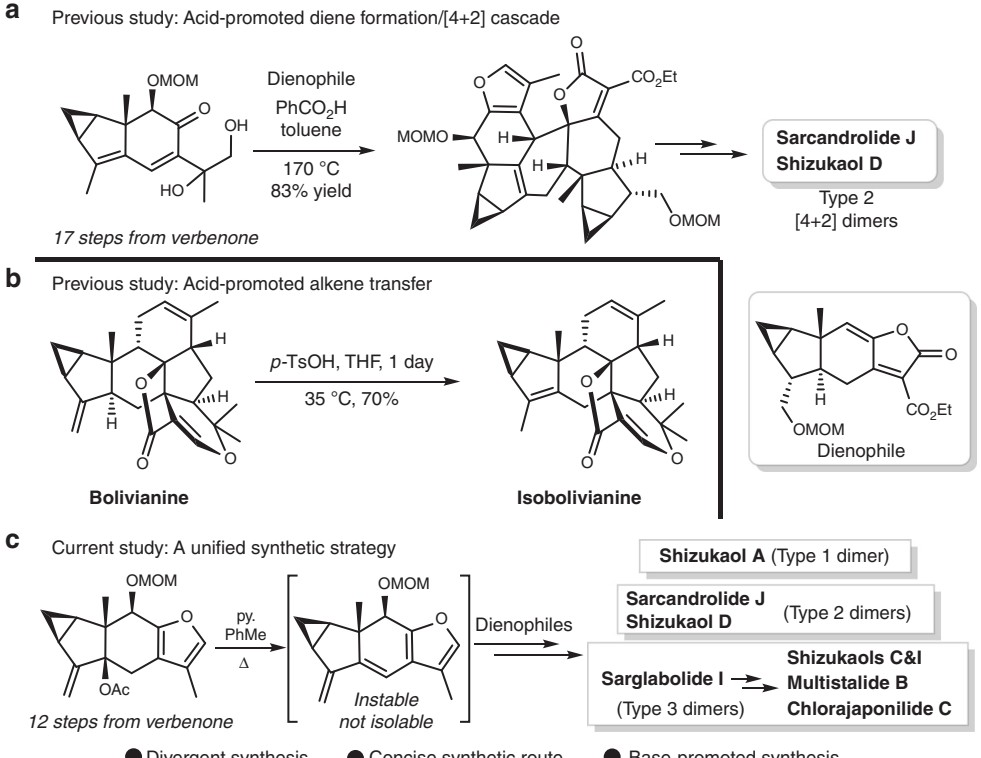

**Fig. 2** Evolution from an acid-promoted cascade to the current unified strategy. **a** The skeleton of type 2 dimeric lindenane sesquiterpenoids was constructed through an acid-promoted diene-formation/[4 + 2] cascade. **b** Bolivianine was transformed to isobolivianine via the acid-promoted *exo/endo* cyclic alkene isomerization. **c** A unified synthetic strategy is developed for the synthesis of three types of dimeric lindenane sesquiterpenoids

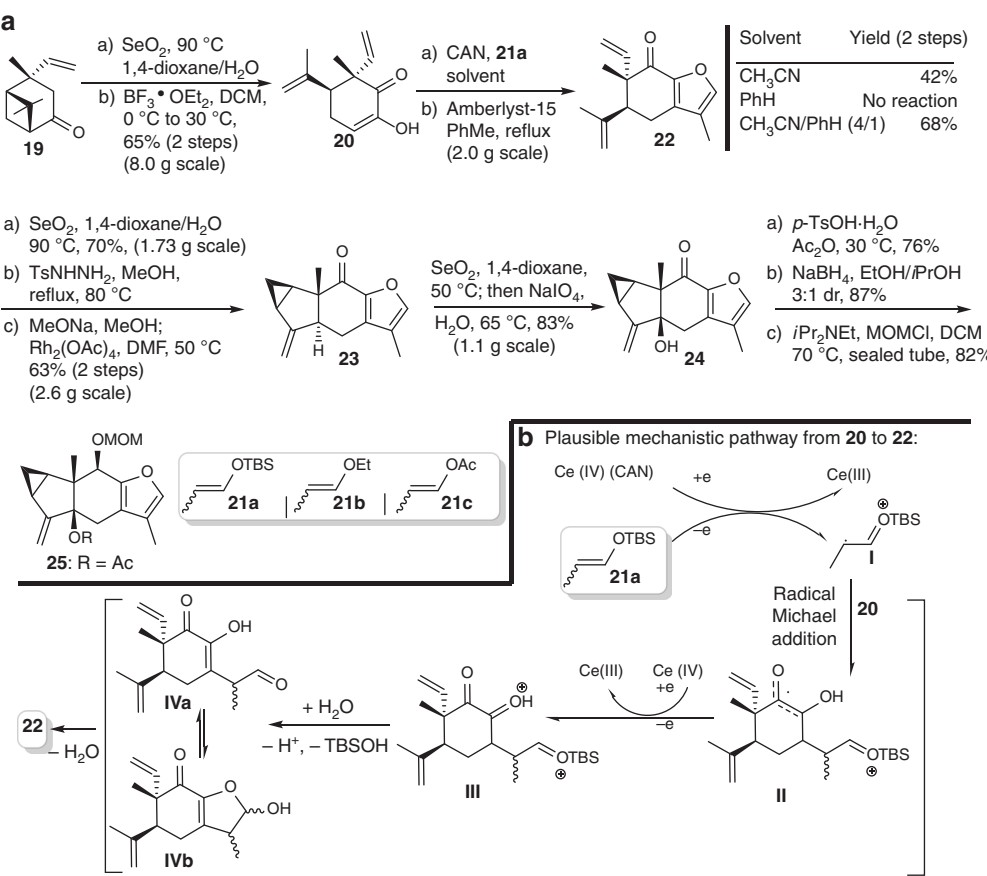

**Fig. 3** Synthesis of the diene precursor **25**. **a** Compound **25** can be synthesized from compound **19** in 11 steps. **b** The mechanism for formation of **22** was proposed. DCM = dichloromethane, DMF = N,N-dimethylformamide, Ac = acetyl, DIPEA = N,N-diisopropylethylamine, MOMCl = methoxymethyl chloride

an unidentified intermediate presumably as allylic selenide[29–31]. Accordingly, sodium periodate was introduced into the reaction mixture, exclusively producing **24** in 83% yield[32–34], while no furan benzylic oxidation product was detected. Finally, acid-promoted acetylation of the tertiary alcohol, diastereoselective reduction of the ketone, and the MOM protection resulted in diene precursor **25**. Mesylation and tosylation of the corresponding tertiary alcohol of compound **24** failed due to low reactivity.

**Synthesis of the dienophile**. We validated our proposed unified synthetic strategy by first aiming at total syntheses of type 3 [4 + 2] dimers. Thus, sarglabolide I (**11**) was chosen as the first target molecule and we began to synthesize the corresponding dienophile partner **36** for sarglabolide I (**11**) from the known compound **26** (Fig. 4a), accessible from verbenone[6,23–25].

Direct dihydroxylation of **26** failed to afford the desired diol **27** using either K₂OsO₄•2H₂O/NMO or Sharpless asymmetric dihydroxylation (SAD) with both AD-mix-α and AD-mix-β. Instead, while compound *epi*-**27** was obtained as the major diastereomer with more than 20:1 dr in all cases (Fig. 4b). We rationalize that these results should come from intrinsic stereochemical control of the rigid chiral skeleton of **26**. Thus, we subjected **26** to both Prévost *trans*-dihydroxylation and sequential halogen hydroxylation-basic hydrolysis, attempting to hydrolyze epoxide **30** to deliver **27**. Compound **30** could be presumably formed through capture of the cyclic halonium ion **28** by hydroxide or carboxylate and subsequent epoxidation from **29** (path a, Fig. 4c). However, only allylic halogen, allylic alcohol and allylic acetate were formed instead, indicating that the undesired base-promoted

deprotonation of **28** proceeded faster to give **31**, transformable to other allylic compounds (path b, Fig. 4c). Thus, we resorted to an alternative route. Ozonolysis of **26**, followed by a cautious workup without adding any reductant, resulted in ketone **32** with minimal epimerization of C5. Although Julia olefination of **32** was feasible to give **26**[10,15], reacting **26** with MOMCH₂SnBu₃/nBuLi at either −78 °C or 0 °C only led to isomerization at C5 to give **33**, serving as a thermodynamic sink with stabilized *cis*-fused 5/6-bicycle. So we considered application of less bulky vinyl organometallic reagent to increase rate of the nucleophilicity over basic isomerization at C5. While vinyl Grignard reagent failed to provide **34** in significant yield, the corresponding cerium reagent afforded **33** as the only byproduct, although organocerium reagents feature high nucleophilicity and weak basicity[35,36]. Fortunately, vinyl lithium promoted the reaction at −78 °C to give **33** and the desired adduct **34** in 34% and 39% yield, respectively. Pleasingly, after several trials, we found the desired nucleophilic reaction would surpass the undesired enolization of **32** much more at higher temperature. In this manner, vinyl lithium was manifested to generate **34** in 61% yield at −10 °C. Ozonolysis of **34** followed by an in situ reduction successfully afforded diol **27**. It was further protected as an acetonide and the glycol-protected ketal was removed under acidic conditions to give compound **35**, which was transformed to **36** over three steps.

**Total syntheses of type 3 [4 + 2] lindenane dimers**. With both **25** and **36** in hand, we next executed the diene-formation/cycloaddition cascade. As shown in Fig. 5, diene precursor **25** was pre-mixed with dienophile **36** and pyridine in a sealed tube upon

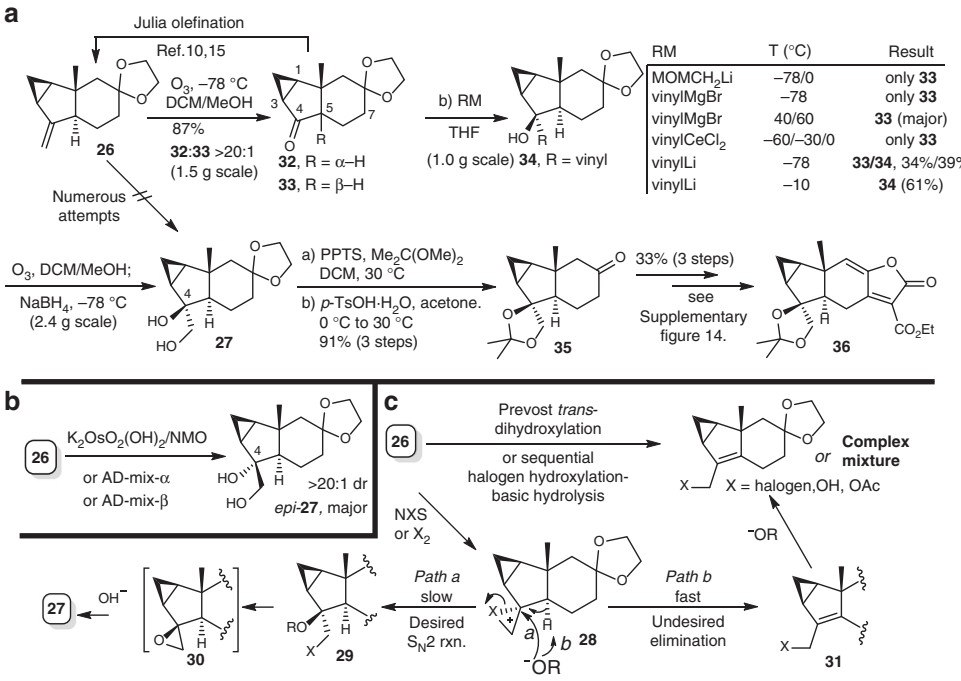

**Fig. 4** Synthesis of the dienophile **36** for type 3 [4 + 2] natural dimers. **a** Compound **36** can be synthesized from **26** in 8 steps. **b** Direct dihydroxylation of **26** fails to afford the desired diol **27**. **c** Both Prévost *trans*-dihydroxylation and sequential halogen hydroxylation-basic hydrolysis fail to afford **27**. PPTS = pyridinium *p*-toluenesulfonate

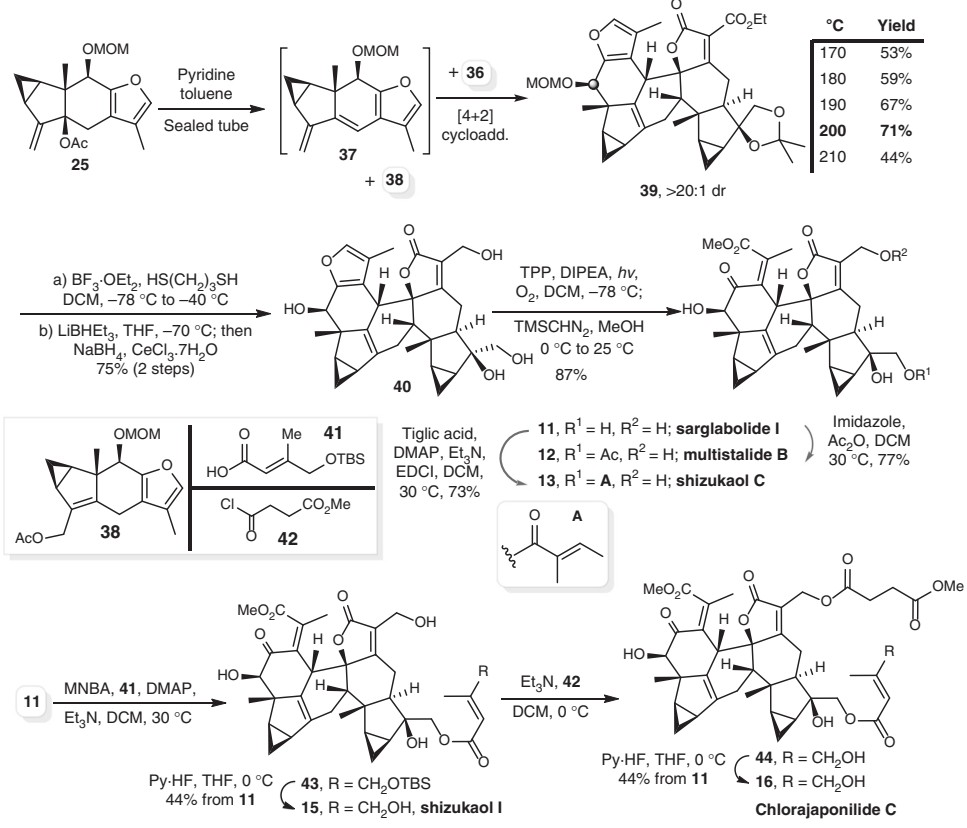

**Fig. 5** Total syntheses of type 3 [4 + 2] lindenane dimers. TPP = 5,10,15,20-tetraphenylporphyrin, DMAP = *N,N*-4-dimethylaminopyridine, EDCI = 1-ethyl-3-(3-dimethylaminopropyl)carbodiimide hydrochloride, MNBA = 2-methyl-6-nitrobenzoic anhydride

the heat. The extrusion of acetate generated diene **37** in situ, which underwent an intermolecular [4 + 2] cycloaddition with **36** to form cycloadduct **39** as the major diastereomer. An optimal 71% yield of **39** was obtained at 200 °C, whereas a Cope rearrangement byproduct (**38**) was also observed, which is stable under this thermal condition. After simultaneous deprotection of MOM ether and acetonide and reduction of the ethyl ester, compound **40** was obtained in 75% yield over two steps. A one-pot protocol afforded sarglabolide I (**11**) in 87% yield through sequential photolytic oxidation of the furan and esterification of the resultant acid[37]. Selective esterifications of **11** using acetic anhydride and tiglic acid afforded multistalide B (**12**) and shizukaol C (**13**), respectively[38,39]. Shizukaol C exhibits potent inhibitory activities against various plant pathogenic fungi[40]. In addition, acylation of sarglabolide I (**11**) offered compound **43** utilizing acid **41**, and the final silyl deprotection afford shizukaol I (**15**)[41]. Moreover, the second O-acylation of **43** gave compound **44**, which went through silyl deprotection to afford chlorajaponilide C (**16**)[42,43]. Notably, this natural product shows very potent *Plasmodium falciparum* growth inhibition with IC$_{50}$ as low as 1.1 nM; it was further tested for mammalian cytotoxicity with WI-38 cell line and show selective index as high as 4900[43]. These findings indicate impressive potential as anti-malarial drug for this compound.

**Total syntheses of [4 + 2] lindenane dimers of types 1 and 2.**
We further applied the unified strategy in synthesis of type 1 and type 2 dimers (Fig. 6a). In the presence of pyridine, mixing compound **25** and previously reported dienophile **45**[6], generated compound **46** as the major diastereomer ( > 20:1 dr) under thermal conditions. In accordance with reported protocols[6], **46** can be transformed into sarcandrolide J (**6**) and shizukaol D (**7**), two [4 + 2] lindenane dimers of type 2[44–47]. Similarly, compound **48** was synthesized in a moderate yield (43%) by heating the mixture of **25** and chloranthalactone A (**47**), synthesized from verbenone as well[17,18,23–25]. After deprotection of the MOM ether and oxidative elaboration of the furan ring, shizukaol A (**4**), a [4 + 2] lindenane dimer of type 1[48], was synthesized. In comparison with the 16% yield of **48** acquired from the acid-mediated

dimerization between **47** and previously known **49** (Fig. 6b)[6], this base-mediated dimerization protocol showcases its satisfactory synthetic efficacy and panoramic tolerance of versatile dienophiles.

## Discussion

In summary, we developed an effective and unified strategy enabling feasible access to all the three types of lindenane sesquiterpenoid [4 + 2] dimers exemplified by syntheses of eight natural products. This natural family composes of more than one hundred dimeric compounds possessing versatile biological activities. Our divergent strategy would undoubtedly lay solid foundation for creation of the analog library and thus facilitate biological evaluations of these lindenane dimers.

## Methods

**General**. All reactions were performed under an argon atmosphere with dry solvents under anhydrous conditions, unless otherwise stated. DCM, DIPA, DIPEA, HMDS, MeCN, DMF, Et$_3$N, and toluene were distilled from calcium hydride under argon; MeOH was distilled from dry magnesium turnings and iodine under argon; THF was distilled from sodium-benzophenone under argon; Ac$_2$O was distilled from K$_2$CO$_3$ after being preprocessed by P$_2$O$_5$. Unless otherwise noted, all the other chemicals were purchased commercially and used without further purification. Flash chromatography was performed using silica gel (200–300 mesh). Thin layer chromatography (TLC) was used for monitoring reactions and visualized by a UV lamp (254 nm and 365 nm), I$_2$ and developing the plates with *p*-anisaldehyde or phosphomolybdic acid. $^1$H and $^{13}$C NMR were recorded on Bruker DRX-400 MHz NMR spectrometer or Bruker 800 MHz NMR spectrometer with TMS as the internal standard and were calibrated using residual undeuterated solvent as an internal reference (CDCl$_3$: $^1$H NMR = 7.26, $^{13}$C NMR = 77.16; C$_6$D$_6$: $^1$H NMR = 7.16, $^{13}$C NMR = 128.06; CD$_3$COCD$_3$: $^1$H NMR = 2.05, $^{13}$C NMR = 29.84; CD$_3$OD: $^1$H NMR = 3.31 and 4.87, $^{13}$C NMR = 49.00; Pyridine-d$_5$: $^1$H NMR = 8.74, 7.58 and 7.22). Abbreviations in $^1$H NMR data are illustrated as follows: s = singlet, d = doublet, t = triplet, dd = doublet of doublet, ddd = doublet of doublet of doublet, dt = doublet of triplet, td = triplet of doublet, m = multiplet, br = broad. Coupling constants (*J*) are reported in Hertz (Hz). Optical rotations were measured at the sodium D line with a 100 mm path length cell, and are reported as follows: [α]$_D^T$, concentration (g/100 mL), and solvent. High-resolution mass spectra (HRMS) were recorded by using Bruker-FT-MS spectrometers. Infrared (IR) spectra were recorded on a NEXUS 670 FT-IR device and are reported in wavenumbers (cm$^{-1}$).

**Experimental data**. For detailed experimental procedures, see Supplementary Methods. For NMR spectra of the synthesized compounds in this article, see

**Fig. 6** Total syntheses of [4 + 2] lindenane dimers of types 1 and 2. **a** Sarcandrolide J, shizukaol D and shizukaol A can be synthesized through the [4 + 2] cycloaddition promoted by base. **b** The acid-mediated cycloaddition between **47** and **49** afforded **48** in inferior yield to that of the base-mediated cycloaddition between **25** and **47**

Supplementary Figs. 26–94. For the comparison of NMR spectra of the natural products and synthetic products, see Supplementary Tables 1–12.

## Data availability

The authors declare that the data supporting the findings of this study are available within the article and its Supplementary Information files. Furthermore, all other data are available from the authors upon reasonable request.

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

## Acknowledgements

We acknowledge financial support from the NSFC (21672153) and the Open Fund of State Key Laboratory of Natural Medicines in China Pharmaceutical University (SKLNMKF201810). We are grateful to Dr. Pengchi Deng in Analytical and Testing Center (SCU) for his help on NMR analysis.

## Author contributions

B.D., Z.H. and B.L. conceived the synthetic strategy and analyzed the results. B.D., Z.H., X.W., T.C., G.S. and S.F. performed experiments. B.L. designed and directed the project and wrote the manuscript.

## Additional information

**Competing interests:** The authors declare no competing interests.

