## [Peer Review File · Nature Communications]

REVIEWERS' COMMENTS:

Reviewer #1 (Remarks to the Author):

In this paper, Liu and coworkers report the divergent syntheses of eight dimeric lindenane sesquiterpenoids, such as shizukaols A, C, D, I, chlorajaponilide C, multistalide B, sarcandrolide J and sarglabolide I, using a unified dimerisation strategy via a base-mediated thermal [4+2] cycloaddition of several types of dienophiles and a common furyl diene, generated in situ. This elegant synthetic strategy may be applied to the synthesis of a large number of the intriguing similar family members and analogs for understanding their relevant biological action and biogenetically synthetic protocol in nature. Moreover, Liu has developed a reliable route for formation of furan species and in-situ assembly of key diene moiety, which made compound 7 as a very easily available common precursor for accomplishment of several natural products. Therefore, I would like to strongly recommend this works to be published in Nature Communications after minor revision on basis of these points as below and the attached documents.

1. It is logical to number relevant molecules according to the sequence shown in the Text, but now the synthetic natural products were to be the last several numbers.
2. In Scheme 2, the conditions of 6 to 7, "i" of iPr₂NEt should be italic form.
3. In Scheme 2, from compounds 5 to 6, "1.1g" should be one more space. I won't specify all the typos, please check the text and supporting materials very seriously and revise them accordingly.
4. In Scheme 3, please revise "a, b" to "α or β" (14, 15, epic-9).
5. In Scheme 3, please use a longer hyphen before the temperature numbers in the reaction conditions, such as "- 78" or "- 10". Please also check the whole text and Supporting materials.
6. In Line 124, "scheme 4" should be "Scheme 4".
7. In Lines 114 and anywhere, please add one more comma before "respectively".
8. In Scheme 4, please correct the format of reaction conditions from 21 to 22.
9. In Scheme 5, it is better to put the structures of 32 and 34 to the schemes.
10. In Scheme 2, please make sure to use half arrow to demonstrate the single electron transfer if have.
11. In all the schemes, please specify the exact temperature for room temperature in the whole text.

Reviewer #2 (Remarks to the Author):

In this manuscript, Liu and his coworkers described an efficient and collective synthesis of eight dimeric lindenane sesquiterpenoids, which are considerably challenging targets. By carefully comparing and analyzing this work and author's previous work on this topic, the reviewer must agree that the authors has made significant efforts to compose new synthetic routes with completely different strategies (i.e. with minimum overlap with their previously published results). Their efforts are clearly impressive: Eight lindenane dimers has been made with various oxidative states in a stunning efficiency. Especially, the seemingly simple oxidative state manipulation on the exo-double bond of these molecules is, actually, very difficult due to the substrate limitation. The authors find a smart way to avoid this issue and successfully introduced the corresponding oxygen atoms in the early stage, thus allowed the production of the so-called type 3 lindenane dimers, for the first time. This very appealing synthesis most definitely should be published in nature communications.

Two questions:

- 1) Was any other single electron oxidant tested for the oxidative coupling of 2 and 3a? (Maybe a related paper could be cited: Chem. Sci., 2012, 3, 3378)
- 2) Was any furan benzylic oxidation product detected in the transformation from 4 to 5 and, from 5 to 6?

Point to Point Response to the Reviewers

Reviewer(s)' Comments to Author:

Reviewer 1

1. It is logical to number relevant molecules according to the sequence shown in the Text, but now the synthetic natural products were to be the last several numbers.

Response: Thanks for the suggestion from this reviewer. We have renumbered the compounds both in main text and supplementary information to ensure the numbers of natural products at the front.

2. In Scheme 2, the conditions of 6 to 7, "i" of iPr₂NEt should be italic form.

Response: We have changed the "i" to "i", and we also have checked the whole text and Supplementary Information and corrected these errors.

3. In Scheme 2, from compounds 5 to 6, "1.1g" should be one more space. I won't specify all the typos, please check the text and supporting materials very seriously and revise them accordingly.

Response: We have checked the text and Supplementary Information to correct these errors.

4. In Scheme 3, please revise "a, b" to "α or β" (14, 15, epic-9).

Response: We have changed "a, b" to "α or β" in Figure 4.

5. In Scheme 3, please use a longer hyphen before the temperature numbers in the reaction conditions, such as "- 78" or "- 10". Please also check the whole text and Supporting materials.

Response: We have checked the text and Supplementary Information to correct these errors.

6. In Line 124, "scheme 4" should be "Scheme 4".

Response: We have changed "scheme 4" to "Figure 5".

7. In Lines 114 and anywhere, please add one more comma before "respectively".

Response: We have checked the text and Supplementary Information to correct the errors accordingly.

8. In Scheme 4, please correct the format of reaction conditions from **21** to **22**.

Response: We have corrected the format of reaction conditions from **39** to **40** in Figure 5.

9. In Scheme 5, it is better to put the structures of **32** and **34** to the schemes.

Response: We have put the structures of **45** and **47** to the schemes in Figure 6.

10. In Scheme 2, please make sure to use half arrow to demonstrate the single electron transfer if have.

Response: We have corrected the wrong arrows to half arrows.

11. In all the schemes, please specify the exact temperature for room temperature in the whole text.

Response: We have specified the exact temperature for room temperature in the whole text and Supplementary Information.

.

Reviewer 2

1. Was any other single electron oxidant tested for the oxidative coupling of **2** and **3a**? (Maybe a related paper could be cited: Chem. Sci., 2012, 3, 3378)

Response: We have tested several single electron oxidants for the oxidative coupling of **20** and **21a**, but led to unsatisfactory results, and this failure has been mentioned in revised manuscript. We have cited the related paper (Chem. Sci., **2012**, 3, 3378) in reference 27.

2. Was any furan benzylic oxidation product detected in the transformation from **4** to **5** and, from **5** to **6**?

Response: No furan benzylic oxidation product was detected in the transformation from **22** to **23** and transformation from **23** to **24**, which has mentioned in the revised

manuscript.

We appreciate all the questions and comments from these reviewers!